# Is It Just Face Blindness? Exploring Developmental Comorbidity in Individuals with Self-Reported Developmental Prosopagnosia

**DOI:** 10.3390/brainsci12020230

**Published:** 2022-02-08

**Authors:** Nanna Svart, Randi Starrfelt

**Affiliations:** Department of Psychology, University of Copenhagen, DK-1353 Copenhagen, Denmark; nannasvart@live.dk

**Keywords:** face recognition, neurodevelopmental disorders, comorbidity, dyslexia, dyscalculia, navigation problems, synesthesia, aphantasia

## Abstract

Developmental prosopagnosia (DP)—or ‘face blindness’—refers to life-long problems with facial recognition in the absence of brain injury. We know that neurodevelopmental disorders tend to co-occur, and this study aims to explore if individuals with self-reported DP also report indications of other neurodevelopmental disorders, deficits, or conditions (developmental comorbidity). In total, 115 individuals with self-reported DP participated in this online cross-sectional survey. Face recognition impairment was measured with a validated self-report instrument. Indications of difficulties with navigation, math, reading, or spelling were measured with a tailored questionnaire using items from published sources. Additional diagnoses were measured with direct questions. We also included open-ended questions about cognitive strengths and difficulties. Results: Overall, 57% reported at minimum one developmental comorbidity of interest, with most reflecting specific cognitive impairment (e.g., in memory or object recognition) rather than diagnostic categories (e.g., ADHD, dyslexia). Interestingly, many participants reported cognitive skills or strengths within the same domains that others reported impairment, indicating a diverse pattern of cognitive strengths and difficulties in this sample. The frequency and diversity of self-reported developmental comorbidity suggests that face recognition could be important to consider in future investigations of neurodevelopmental comorbidity patterns.

## 1. Introduction

Without the ability to recognize faces, one would live in a world of strangers. This is the reality of people suffering from developmental prosopagnosia (DP), a neurodevelopmental disorder of face recognition. The hallmark of this disorder is severely impaired face recognition in the absence of brain injury, intellectual impairment, or significantly impaired vision [1]. Consequences of DP vary in severity but may include anxiety and compromised social and occupational functioning [2,3,4]. While acquired prosopagnosia following brain injury to the occipito-temporal or anterior temporal regions [5,6] has been studied for many decades, DP did not receive notable research interest until this millennium [7,8]. This is intriguing, as estimates suggest that 2–2.5% of the population suffer from DP [9,10], but see [11] for a discussion. In comparison, more known disorders, like obsessive-compulsive disorder (OCD) or attention deficit hyperactivity disorder (ADHD), have a (lifetime) prevalence of 1.3% [12] and 2.8–4.4% [13,14], respectively. DP research has primarily been concerned with mechanistic or cognitive interpretations of the disorder, and in particular the question of whether individuals with DP also have abnormalities in other domains [15,16,17,18]. Specifically, it has been debated whether object agnosia and reading difficulties could be associated with DP, and this has been linked to discussions about the functional organization of high-level vision, e.g., [19,20,21,22,23,24], see [25] and accompanying commentaries for review. However, the broader question of whether face recognition deficits might be overlapping with other neurodevelopmental disorders or conditions has received less attention, and here we aim to take the initial steps to explore this question.

It is well-known that neurodevelopmental disorders tend to cluster and co-occur [16,26,27,28], e.g., comorbid dyslexia and ADHD [29,30], but the co-occurrence of DP or face recognition problems with other diagnostic categories is largely unknown. Anecdotal evidence and some initial studies do suggest some overlap between DP and some other specific cognitive deficits/disorders with a developmental origin, e.g., navigation problems (developmental topographical disorientation) e.g., [31] and dyslexia e.g., [32], but if and how DP or face recognition deficits fits into a larger ‘comorbidity space’ of neurodevelopmental disorders is at present unknown.

The exception is the relation to autism spectrum disorder (ASD), where observations of face recognition deficits are common [33,34,35]. However, ASD often presents with severe social and intellectual deficits not found in DP, which is why ASD is one of the most common exclusion criteria from DP studies [1]. In addition, ASD can co-occur with a range of other neurodevelopmental disorders independently of face processing deficits. For these reasons, the potential overlap between ASD and DP is not considered in the present study, and participants with ASD were excluded from participation.

One reason why neurodevelopmental comorbidity in DP has not been described could be the lack of a gold standard to identify DP. Studies investigating DP have typically applied few and broad exclusion criteria, which also vary substantially between studies [1,11]. The majority of studies investigating DP include participants performing 1.7–2 standard deviations (SD) lower than a control mean of one or more face processing tests [11], e.g., The Cambridge Face Memory Test (CFMT) [36], and the most common exclusion criteria are ASD, brain damage, and/or neurological disorder [21,25,37,38,39]. Thus, with the exception of ASD, potential co-occurring neurodevelopmental disorders are typically not part of inclusion or exclusion criteria in studies of DP.

In a first attempt to address the question of neurodevelopmental comorbidity in DP, we took an exploratory and broad approach to allow for potentially valuable unexpected findings [40], an approach which of course also presents with a number of study design challenges that may limit the generalizability of the results.

We asked whether *self-reported developmental prosopagnosia (s-DP)* is likely to co-occur with other neurodevelopmental disorders or conditions (comorbidities), and if so, to what degree. To this aim, we asked how people with s-DP rate their navigation, reading, spelling, and mathematical abilities, which all represent potentially related cognitive abilities/impairments [15,41]. In addition, using directed content analysis, which is a qualitative descriptive approach, we analyzed the respondents’ own descriptions of their difficulties as well as potential strengths [42]. Because of the lack of previous knowledge to guide hypotheses, we chose a coarse-grained and exploratory approach, and thus there were no formal statistical hypotheses to be tested. We were, however, particularly interested in the relationship between the severity of self-reported face recognition deficits and presence of comorbidities and exploring the relationship between ratings of face recognition, and items regarding navigation, reading, spelling, and mathematical abilities. In addition, we were interested to see whether open-ended questions could provide indications of possible comorbidities that have not previously been explored in relation to DP. We would like to acknowledge from the start that the participants in this study were self-referred and their face recognition impairment was assessed by a questionnaire only. Thus, we cannot determine whether our participants would qualify for an actual diagnosis of DP based on the present data, as this demands additional objective evidence of impairment on tests of face recognition [11,43]. We do, however, still think our findings can provide a starting point for future investigations of the relationship between face recognition impairment and developmental disorders.

## 2. Material and Methods

### 2.1. Which Neurodevelopmental Deficits and Conditions Could Be Relevant to Assess?

While we wanted to take a broad and open approach, we also used existing literature to guide the inclusion of relevant types of developmental disorders or deficits previously investigated in relation to face processing or DP as a starting point. These included navigation problems/developmental topographical disorientation [15,31,44,45,46], object agnosia e.g., [25], and reading problems/dyslexia [21,23,32,47,48,49,50,51]. As a novelty, we included dyscalculia, a developmental deficit in number processing, as this is commonly associated with dyslexia [52,53,54]. Along the same vein, we included ADHD, which is reported in one DP case study [55], and which also tends to co-occur with dyslexia [29,30]. Face processing deficits have also been indicated in the neurodevelopmental condition synesthesia, which is characterized by a merging of senses such as seeing letters in colors [56,57]. Synesthesia is an atypical developmental condition, but not a disorder per se as it is not associated with deficits in or loss of function. Rather, it has been suggested to be related to creativity [58] and enhanced memory [59]. Lastly, memory problems in a developmental variant were found relevant to investigate for two reasons: First, because DP in many cases is found to reflect problems with face memory [60,61]. Secondly, because of observations of aphantasia, a deficit in voluntary visual mental imagery [45,62,63] in DP. Aphantasia has been associated with severely deficient autobiographical memory [64], which is a deficit in re-experiencing personal events [65]. This raises the question as to whether severely deficient autobiographical memory could also be linked to DP.

In sum, and in alphabetical order, aphantasia, attention-deficit/hyperactivity disorder, dyscalculia, dyslexia, memory problems, navigation problems, object agnosia, and synesthesia were considered relevant conditions to investigate in DP. For practical reasons, they are collectively referred to here as *developmental comorbidity*.

### 2.2. Participants

This cross-sectional survey study used convenience sampling. In total, 148 respondents filled out the Difficulties and Abilities in Developmental Prosopagnosia Questionnaire (described in Section 2.4). Exclusion criteria were (1) incomplete responses (*n =* 13); (2) reports of neurological disorder or brain injury (*n =* 4); (3) not reporting normal vision (*n =* 4); and (4) neurodevelopmental or psychiatric disorders that can cause impairment in intellectual or social functioning (*n =* 12), e.g., ASD [8,34] and schizophrenia [66,67]. Overall, 115 participants with s-DP were included. Participants were encouraged to participate if they experienced difficulties recognizing familiar faces in their everyday life and thus were suspected to suffer from DP. Thirty-four (29.6%) participants also reported previously being diagnosed with DP by a health professional or researcher. The participants were represented broadly in all age intervals from 18–24 years to 75 years or older (Table 1). Furthermore, 33% of the participants fell within the median interval of 35–44 years of age. All included participants reported normal or corrected-to-normal vision. Participants were not asked to indicate their gender for data protection reasons. Demographic data are reported in Table 2.

### 2.3. Data Collection

The survey was distributed online using the software SurveyXact By Ramboll [68]. The participants were recruited via the social networking platform Facebook, where the survey was posted in three groups (two international and one Norwegian group) with relevance to prosopagnosia. The survey was written in English and was available for six weeks.

Informed consent was obtained from all participants prior to their participation, and they were informed that all collected data would be anonymous. The study was conducted in adherence to the Declaration of Helsinki [69], and received ethical approval by the Institutional Ethical Review Board, University of Copenhagen, Department of Psychology (Project ID: IP-IRB/02042020).

### 2.4. Measures

#### 2.4.1. The Difficulties and Abilities in Developmental Prosopagnosia Questionnaire

To measure indications of potential developmental comorbidity and abilities in s-DP, we constructed the Difficulties and Abilities in Developmental Prosopagnosia Questionnaire. It comprised items from published questionnaires aimed at adult participants, as well as items specifically formulated for this study (see Section 2.4.2, Section 2.4.3, Section 2.4.4, Section 2.4.5 and Section 2.4.6). The full questionnaire is available in the Appendix A. In addition to the items from formal questionnaires, two items asked directly about specific other diagnoses or comorbidity with categorical response options (yes/no), and a free-text field for adding other diagnoses. The first of these items concerns primarily well-known developmental disorders of interest: ‘*Do you have other diagnoses? Particularly related to vision, mental illness, learning difficulties, sensory disorders, developmental disorders, motor disorders, etc.*’ with the response options ADHD, ASD, dyslexia, dyscalculia, and other (free text). The second item was assessing self-reports of other, perhaps less conventional conditions of interest: *‘Do you have any of the following conditions related to cognitive strengths or difficulties?*’ with the response options aphantasia, synesthesia, object agnosia, memory problems, and other (free text). Finally, our questionnaire included two open-ended questions regarding cognitive strengths and difficulties, respectively, to allow participants to describe this in their own words (see Section 2.4.6).

It takes approximately 15–20 min to fill out the Difficulties and Abilities in Developmental Prosopagnosia Questionnaire, and it was kept short due to the online distribution format [70].

#### 2.4.2. The 20-Item Prosopagnosia Index (PI20)

To assess self-reported face recognition in the sample of s-DP, the full PI20 was included in the Difficulties and Abilities in Developmental Prosopagnosia Questionnaire. The PI20 is a widely used 20-item self-report measure of face recognition developed to identify and assess severity of prosopagnosia, [71]. The PI20 uses a 5-point Likert-type scale (1 = strongly agree to 5 = strongly disagree) to score agreement with statements about face recognition experiences, e.g., ‘My face recognition ability is worse than most people’ and ‘At family gatherings, I sometimes confuse individual family members’ [71] (p. 4). The psychometric properties of the PI20 are good: In the original validation study by Shah et al. [71] it was shown to have very high internal consistency (Chronbach’s α = 0.96), and good convergent validity as it correlates well with objective measures of face recognition, e.g., CFMT (*r* = −0.68, *p* < 0.0001). The maximum score is 100, and higher scores indicate more difficulty recognizing faces. Scores of 65–74, 75–84, and 85–100 can broadly indicate mild, moderate, and severe DP, respectively [71].

#### 2.4.3. Items from the Wayfinding Questionnaire

To investigate navigation abilities, the Difficulties and Abilities in Developmental Prosopagnosia Questionnaire included five items from the Wayfinding Questionnaire [72,73], which uses a 7-point Likert-type scale (1 = Not at all applicable to me, 2 = Almost never applicable to me, 3 = Rarely applicable to me, 4 = Sometimes applicable to me, 5 = Sometimes applicable to me, 6 = Almost always applicable to me, and 7 = Fully applicable to me.). Higher scores indicate better wayfinding ability. The Wayfinding Questionnaire was originally developed for screening for navigation-related complaints in stroke patients and was applied here as a substantial number of individuals with DP have voiced similar complaints [44,45]. De Rooij and colleagues [73] (p. 1051) define a score of ≤3 on an item as indicating clinically relevant complaints. The Wayfinding Questionnaire has been validated in both healthy participants (*N* = 356) [72] and stroke patients (*n* = 78) [73]. The stroke patients with low scores on the Wayfinding Questionnaire performed worse on objective measures of navigation ability in a virtual reality setting, suggesting good concurrent validity [73,74]. Four of the five chosen items have been applied previously by Rice et al. [75], and the fifth item was included as it concerned indoor navigation. The five items all had satisfactory loadings on the Wayfinding Questionnaire subscale ‘Navigation and Orientation’ (0.631–0.841) [72].

#### 2.4.4. Items from the Adult Reading History Questionnaire

The Adult Reading History Questionnaire (revised) is a 23-item self-report questionnaire, one of the few designed to identify a history of reading difficulties indicating dyslexia in adults [76]. As mentioned, face recognition difficulties have been identified in dyslexia [32,48,49,50], and a common etiology for dyslexia and DP has been suggested [28,77]. Even though normal reading skills have been demonstrated in DP [21,22,23,78], the relationship between the disorders is not fully understood, and hence the Adult Reading History Questionnaire was included in the current study. The Adult Reading History Questionnaire has good psychometric properties [76,79], including good internal consistency (Chronbach’s α = 0.94 and 0.92), test-retest reliability (*r* = 0.87 and 0.84), and construct validity (*r* = 0.57–70 between the Adult Reading History Questionnaire and reading measures) [76]. Furthermore, the Adult Reading History Questionnaire has predicted the occurrence of dyslexia reasonably well [76].

Responses to the Adult Reading History Questionnaire are given on a 5-point Likert-scale, with higher scores corresponding to more difficulty. The original range is 0–4, but in the current study it was adapted to 1–5 to provide consistency amongst the majority of the scales [70]. The four items with the highest loadings (0.920–0.794) on the subscale ‘Dyslexia Symptoms’ were chosen to be included in our questionnaire [79]. These four items were concerned with literacy skills in elementary school, and a fifth item was included to represent current reading ability. As we only used a subset of the items from this questionnaire, the psychometric properties of the included items are unknown.

#### 2.4.5. Items Concerning Mathematical Ability

Three items were constructed as no self-report measures of mathematical ability and dyscalculia for adults exist [80,81]. Schelke et al. [80] suggested the math subscale from the Colorado Learning Difficulty Questionnaire [82], originally a parent-report questionnaire, as a candidate for investigating dyscalculia retrospectively in adults. Two of these items were rephrased to target an adult population. One item (‘Are/were your child worse at math than at reading and spelling?’) was discarded as its content was misleading due to the high co-occurrence between dyscalculia and dyslexia [83]. Instead, a third item was formulated concerning estimation of quantities (subitizing), which is central to dyscalculia [84,85]. These three items were graded with a 5-point Likert-type scale (1 = strongly agree to 5 = strongly disagree).

#### 2.4.6. Open-Ended Questions Regarding Difficulties and Abilities

The Difficulties and Abilities in Developmental Prosopagnosia Questionnaire also contains two guided open-ended questions aimed at a less constrained and suggestive assessment of the abilities and difficulties experienced by participants with s-DP in their own words. Aligned with the explorative nature of our questionnaire, this provides an opportunity for non-expected findings or responses [86]. The questions were phrased ‘*Are there things that you are particularly good at, where you experience performing better than others?*’ and ‘*Are there things or situations, where you experience performing worse than others?*’. The questions were provided with a guiding text, e.g., ‘*Please consider and elaborate on any difficulties you might have, for example within the areas of math, memory, navigation, visualizing images or objects, literacy, sports or tone deafness*’.

## 3. Data Analysis

### 3.1. Statistical Analysis

The statistical analyses were conducted using IBM SPSS, version 27.0 [87] with a significance level of *p* ≤ 0.05.

#### 3.1.1. The Effect of Prosopagnosia Severity on the Presence of Developmental Comorbidity

A Chi-square test of independence was used to analyze whether the presence of any binary-measured developmental comorbidity (ADHD, aphantasia, dyscalculia, dyslexia, memory problems, object agnosia, or synesthesia) was independent of severity of self-reported face recognition impairment (PI20 total scores) within the sample of participants with s-DP [88,89]. The guiding criteria from the PI20 was used as cutoffs for dividing the severity groups into mild DP with total scores 65–74, moderate DP with total scores 75–84, and severe DP with total scores 85–100 [71].

#### 3.1.2. Exploratory Factor Analysis of the Difficulties and Abilities in Developmental Prosopagnosia Questionnaire

An exploratory factor analysis was performed to investigate the latent variables underlying the responses of the sample of individuals with s-DP (*N* = 115) to the PI20, the Wayfinding Questionnaire, the Adult Reading History Questionnaire, and the mathematical ability items. In other words, the interest is the underlying structure and relationship between self-reported face recognition ability and navigation, reading/spelling, and mathematical ability. A reasonable assumption could be that a four-factor structure would emerge. It has been demonstrated that the PI20 has one factor [71], and the Wayfinding Questionnaire, the Adult Reading History Questionnaire, and the math items would be expected to represent different abilities, and hence different factors. The comorbidity patterns described above [31,32,52], e.g., co-occurring developmental reading and mathematical difficulties, could allow for speculations of correlation between these measures, potentially resulting in three factors. However, in both cases the knowledge is still too limited to allow for precise predictions of the factor outcome, and thus an exploratory approach was taken.

Fabrigar et al. [90] argue that principal axis factoring is the most adequate choice of factor extraction method for exploratory factor analysis when data are not normally distributed. This is typically the case for ordinal data obtained using a Likert-type scale [91], which also applies to the data in the current sample when considering general guidelines for determining the distribution of the data [92] (p. 61).

Rotation in exploratory factor analysis is performed to simplify and clarify data structure as well as improving the interpretation of the factors. An oblique type of rotation (direct oblimin in SPSS) was chosen to allow the factors to correlate [93]. The number of factors to retain was based on the Kaiser criterion, stating that factors with an eigenvalue of more than 1.0 should be retained and checked for agreement with the scree test [93] and qualitative interpretation to find the most meaningful number of factors to extract. Kaiser–Mayer–Olkin measure and Bartlett’s sphericity were applied as measures of the factorability of the dataset, and 0.6 was used as cutoff for an acceptable sampling adequacy [94] (p. 668). Factor loadings of above 0.32 or below −0.32 were regarded as reflecting interpretable and meaningful relationships between an item and a factor, as a factor loading of this size corresponds to approximately 10% of the overlapping variance [94].

### 3.2. Qualitative Analysis

Directed content analysis, a deductive approach, was applied to the responses of the open-ended questions to investigate the frequency and variance in themes regarding abilities and difficulties, and thus provide a descriptive summary of what was reported [42]. A mainly deductive qualitative approach was found appropriate considering the questions and the more quantitative framework of the whole Difficulties and Abilities in Developmental Prosopagnosia Questionnaire, which the open-ended questions were a part of.

Initial coding categories and operational definitions were developed for the statements concerning difficulties based on the developmental comorbidities (except for synesthesia, which is not considered a disorder as there is no loss of function) [95]. First version codes were based on the examples provided in the guiding text, e.g., ‘(…) *mathematical abilities, memory, navigation, visualizing images or objects, literacy, sports or musical talents*’. When appropriate and reflected in the data, the first version codes were created in pairs reflecting a strength and a difficulty, e.g., Good Memory vs. Memory Problems. N.S. (first author) coded the first version codes and recorded meaningful information provided about experienced abilities and difficulties that was not captured by these first version codes. Secondly, new categories, e.g., Skills In Visual Arts, were created in an inductive fashion to fit the data that could not be coded with existing codes but still seemed to reflect meaningful themes in the data [95]. Thus, when a meaningful amount of description of either difficulty or ability was present in the data and was not captured by the first version codes, a new code category was titled and described in the coding manual. Meaningful in this regard entails number of descriptions, richness of description, magnitude of described impact on daily life, and potential relatedness to the cluster of developmental comorbidities.

Recoding and revisiting the coding categories was repeated as a part of an iterative process. A final coding manual was written and discussed with a second independent coder (S.B.J., a student assistant) and revised accordingly (the coding manual is available at OSF, see Appendix A). During the second coder’s coding, doubts and disagreements were discussed continuously between the two raters, and necessary revisions to the codes and the coding manual were made to ensure objectivity and credibility [42]. The interrater reliability between N.S. and S.B.J. was 0.84, measured as a weighted Kappa coefficient. R.S. (second author) served as an expert coder and decided between significant disagreements in the coding between the two initial coders. For minor discrepancies in the coding, N.S. was considered the main coder, unless S.B.J. and N.S. agreed that S.B.J.’s coding adhered more closely to the coding manual for the particular discrepancy. Data were stored and coded using NVivo software, which was also used for the calculation of the Kappa coefficient [96].

## 4. Results

The 115 participants’ responses to the Difficulties and Abilities in Developmental Prosopagnosia Questionnaire showed that they generally experienced problems with face recognition in their everyday life. The average total PI20 score was 83.8 (SD = 8.5), which is close to the suggested cutoff for severe DP, which is 85 [71]. For the severity groups based on the PI20 [71], 15 participants scored within the mild DP group, 42 participants scored within the moderate DP group, and 58 scored within the severe DP group. Two individuals had PI20 total scores below the mild cutoff of 65 and were included in the group with the lowest scores, thus creating a group for mild s-DP and individuals with self-reported face recognition difficulties scoring under the mild cutoff. The total scores of these individuals were 61 and 48, respectively. For the items from other scales, the median (range) was as follows: Wayfinding Questionnaire: 19 (30); Adult Reading History Questionnaire: 6 (17); and mathematical ability items: 6 (12).

### 4.1. Amount of Self-Reported Developmental Comorbidity

The descriptive results for self-reported developmental comorbidity are reported in Table 3 and Table 4.

### 4.2. Prosopagnosia Severity and Developmental Comorbidity

The Chi-square test of independence did not reveal a significant relationship between presence of developmental comorbidity (navigation problems were not included in this analysis) and severity of self-reported face recognition impairment (PI20 total scores) within the sample $\left(p=0.79\right)$. We report this while being aware that p-values should be interpreted with great caution in explorative studies [40]. For the severity groups, the cases of developmental comorbidity were distributed as outlined in Table 5.

This suggests that the presence of developmental comorbidity is independent of severity of self-reported face recognition impairment [71] in this sample.

### 4.3. Factor Structure of the Difficulties and Abilities in Developmental Prosopagnosia Questionnaire

An exploratory factor analysis extracting factors with principal axis factoring, using an oblique rotation, was applied to assess the underlying structure of the latent variables explaining the covariance amongst the responses to our questionnaire from the 115 included participants with s-DP. The data demonstrated good factorability: The Kaiser–Mayer–Olkin measure of sampling adequacy was 0.696 and above the accepted value of 0.6 [94] (p. 668). Additionally, all items except for four (0.457–0.476) met the cutoff of above 0.50 for individual measures of sampling adequacy when inspecting the diagonal element of the anti-image correlation matrix in the preliminary analysis of the data’s fit for exploratory factor analysis [97]. As it was close to the cutoff, the items were retained. Bartlett’s test of sphericity was significant $\left(\chi^{2}\;\left(528\right)\;=1757.6,\;p<0.001\right).$ At first, solutions with up to 10 factors emerged with initial eigenvalues >1, together explaining 55.9% of the extracted variance. Investigating the scree plot (Figure 1) suggested retaining five factors due to the elbow on the curve [93,98].

However, the theoretical a priori factor structure was assumed to have four factors or less if the factors were correlated, and the five-factor structure was not considered qualitatively meaningful and less stable than the four-factor structure when assessed (the five-factor solution is available at the OSF site associated with the project). Costello and Osborne [93] (p. 3) argue that interpretability and stability of the factor structure are also important to consider when analyzing the number of factors to extract. This was thus judged by comparing the item loading tables (pattern matrix in SPSS), which favored a four-factor structure due to interpretability, as the five-factor solution divided the PI20 into two factors with cross-loading items, which is diverging with previous validation studies using exploratory factor analysis reporting a single factor structure of the PI20 [71,99]. Thus, four factors were extracted, and they accounted for 39.9% of the variance. The factor loadings of the items in our questionnaire on the four-factor structure are displayed in Table 6.

The first factor (Developmental Prosopagnosia Symptoms) consisted of 14 items loading above 0.32 on the factor, assessing different aspects of impaired face recognition, such as negative feelings associated with not recognizing someone (e.g., Item 11 and 16 from the PI20), not recognizing familiar people (e.g., Item 4 and 20 from the PI20), and coping strategies (e.g., Item 7 and 10 from the PI20). All five navigation items had strong loadings on the second factor (Navigation and Orientation). The five items assessing reading and spelling ability, primarily in elementary school, collectively loaded on the third factor (History of Reading/Spelling Difficulty). The three items concerning mathematical ability in childhood (e.g., mathematical ability Item 1 and 2) and current difficulties assessing quantity (e.g., mathematical ability Item 3) loaded strongly on the fourth factor (History of Difficulties with Math).

Additionally, Item 3 and 12 from the PI20 loaded on Factor 3, and Item 19 from the PI20 loaded on the fourth factor, which did not correspond with the content of the rest of the items within these factors. However, these loadings were quite small, e.g., just around −0.32. Items 2, 8, and 9 from the PI20 did not load above 0.32 or below −0.32 on any factor.

All factors had at minimum three strong factor loadings (above 0.5 or below -0.5), and all factors consisted of more than three items, suggesting a stable solution [93]. Factor 1 and 3 had a weak correlation of r=−0.168, suggesting that an oblique rotation was appropriate [97]. Additionally, the residuals indicated a good fit with 35% non-redundant residuals (e.g., less than 50% of residuals had values greater than 0.05 in the Reproduced Correlation Matrix) [97].

### 4.4. Qualitative Categories from the Open-Ended Questions

Applying directed content analysis to the open-ended questions in the DAP-Q allowed for more detailed analysis of self-reports of the strengths and difficulties experienced by the participants. The coding categories representing the difficulties and strengths are presented in Table 7, with coding frequencies and a quote serving as an example of the category. Table 7 provides an overview of the diversity within the participants’ qualitative reports of their strengths and difficulties. A range of categories reflects domains where both impairments and strengths are reported (by different individuals). This applies to visual imagery, mathematical abilities, reading/spelling, memory, musicality, sports skills, and navigation, where, interestingly, some participants reported severe problems while others reported excelling. For some domains, e.g., imagery, reading/spelling, and navigation, more participants reported either strengths or difficulties, while for other domains (e.g., math, musicality) representations of strengths and difficulties were more evenly distributed. For some categories, only impairment (object recognition, motor coordination, inattention), or abilities/strengths (synesthesia, social interaction, visual arts) were represented in the qualitative categories.

## 5. Discussion

### 5.1. Discussion of Results

#### 5.1.1. Summary of Results

This study was designed to investigate which types of developmental comorbidity, as well as subjective difficulties and strengths, are reported by participants with s-DP. First, most participants in the study reported face recognition problems on the PI20 compatible with prosopagnosia, although we note again that we did not ascertain this in any other way. Two participants scored lower than the suggested range for DP, but we decided to keep their responses in the analysis as they did self-report significant problems with faces. Fifty-eight participants scored within the most severe group, as defined by Shah et al. [71].

More than half of the sample reported the presence of at least one of the predefined conditions of interest (aphantasia, ADHD, dyscalculia, dyslexia, memory problems, object agnosia, and synesthesia). Relatively few participants reported dyslexia, dyscalculia, and ADHD. Synesthesia was reported by 10.4%. Aphantasia was reported by ~20%, and memory problems and object agnosia were reported by almost a fourth of the sample of individuals with s-DP. In total, 25% reported indications of clinically relevant navigation complaints on all of the selected Wayfinding Questionnaire items. In addition, the presence of developmental comorbidity was observed to be independent of severity of s-DP (PI20 total score) when the participants were divided into a mild, moderate, or severe group.

Finally, an exploratory factor analysis revealed a stable four-factor structure corresponding to the independent scales included, with a very weak correlation between the factors of Developmental Prosopagnosia Symptoms and History of Reading/Spelling Difficulty. The stability of the factor structure serves as an indication of validity of the composition of items, allowing for interpretation and analysis of the data from our questionnaire.

When asked to describe their difficulties and strengths, e.g., in which domains they experience performing better or worse relative to others, a considerable number of participants report difficulties in the categories Navigation Problems, Difficulties with Math, Aphantasia, Memory Problems, and Difficulties with Sports. For strengths, the statements most frequently match the coding categories Good Memory, Reading and Spelling Abilities, Musicality, and Mathematical Abilities. For Aphantasia and Navigation Problems, more participants reported impairment, while comparatively fewer reported performing well in these domains. For Reading and Spelling and Memory, more participants with s-DP reported it as a strength rather than a difficulty, even though many also described memory problems. For skills within math, musicality, and sports, there were a number of reports of both strengths and difficulties.

#### 5.1.2. Discussion of Quantitative Results

The exploratory factor analysis showed a clear and stable four-factor structure of the responses to our questionnaire, largely separating the different (dis)abilities in this sample of s-DP into self-reports of face recognition ability (PI20), navigation (the Wayfinding Questionnaire), reading (the Adult Reading History Questionnaire), and mathematical ability.

However, the factor structure also revealed some interesting aspects of the PI20, where some items did not load on the factor of Developmental Prosopagnosia Symptoms (the first factor) (Item 3, 12, and 19), but instead on the third or the fourth factor, and some items did not load sufficiently on any factor (Items 2, 8, and 9) [93]. Common aspects of these poorly performing PI20 items were either being positively worded (Items 3, 8, 9, and 19) or potentially not distinguishing adequately between the non-DP and DP. Exemplifying the latter, Item 3 seemed to reflect a statement that is characteristic to normal face recognition as well, e.g., typical individuals would also be expected to answer that it is easier to recognize people with distinctive facial features. Thus, although the original validation study of the PI20 revealed a strong single factor [71], both when including controls and DP-suspects and analyzing DPs in isolation, our results indicate that some items could require re-evaluation.

Furthermore, Factor 1 (Developmental Prosopagnosia Symptoms) and Factor 3 (History of Reading/Spelling Difficulty) correlated weakly (*r =* −0.168), suggesting a slight association between reading and face recognition ability in this sample. On one hand, it could simply be due to the PI20 Items 3 and 12 which loaded weakly on the History of Reading/Spelling Difficulty factor. On the other hand, individuals with dyslexia can have impaired face processing, e.g., [49], and perhaps some participants with both were included in the present sample.

Regarding the fourth factor, History of Difficulties with Math, it is worth noting that these items showed a strong factor with convincing factor loadings (0.653–816), even though that there were only three items in total [93]. These were the items we were least sure of when constructing the scale, but the factor analysis seems to indicate that these items might be useful in a future attempt to construct a self-report scale for adult dyscalculia.

#### 5.1.3. Comparing the Quantitative and Qualitative Results

Both the quantitative and the qualitative analyses provided insights into the difficulties and developmental comorbidities that the participants experienced, and the qualitative data further allowed us to understand their self-reported strengths as well. Hence, we will discuss the quantitative and qualitative reports of difficulties and strengths in s-DP in relation to each other and the available literature. To provide a structure for this section, we divided it into cognitive domains/disorders which are presented in alphabetical order.

*Attention Deficit Hyperactivity Disorder.* Seven (6.1%) s-DP participants reported ADHD as a comorbid disorder, which is within the range of self-report of ADHD in college students of between 2–8% [100]. Within the qualitative data there were only two descriptions matching Problems with Inattentiveness and Impulsivity or Hyperactivity. ADHD-like symptoms are generally not mentioned in the literature on DP, e.g., in these overviews [25,45], and to our knowledge there is only one published report of comorbid DP and ADHD [55]. There is, however, high comorbidity between ADHD and reading difficulties [29,30], and as mentioned, dyslexia can present with face processing deficits e.g., [49], which may perhaps be part of the explanation for why a few participants in this sample of s-DP reported ADHD.

*Mathematical Abilities or Difficulties.* Both mathematical ability and difficulties with math were reported at fairly high levels relative to the rest of the qualitative codes. However, only four (3.5%) participants reported dyscalculia in the close-ended questions (see Table 4). To our knowledge, no prevalence studies of adults exist, but the impairment can continue into adulthood [52]. Generally, the struggles described in the qualitative data seemed to revolve around mental math and basic arithmetic operations—or simply feeling ‘below average’. The mathematical skills are broadly or very simply described, but some participants mention academic achievements. The discrepancy between the qualitative and quantitative data could suggest that the difficulties with math that some experience might still fall within a normal range, or affect abilities not typically considered as dyscalculia. We are unaware of any investigations of face processing and dyscalculia, and generally self-reported dyscalculia in adults has received little research attention [80], hence the literature is sparse. This may partly be because it is difficult, if not impossible, to identify dyscalculia without rigorous testing, as there is no current self-report measure available for adults [80,81].

*Memory and Aphantasia*. Lack of mental imagery, which we here interpret as possibly reflecting Aphantasia, was reported frequently both in the qualitative and quantitative data, by 24 participants (20.9%) in both cases. Interestingly, Good Visual Mental Imagery was also reported, although not as frequently (*n* = 7).

Given the suggested link between aphantasia and severely deficient autobiographical memory, memory deficits were also of interest [64,101]. Memory problems were reported by 27 (23.5%), hence by almost a fourth of the participants, and interestingly, a co-occurrence of self-reported aphantasia and memory problems was seen in 11 (9.6%) participants. This is in line with another survey study, where participants (*N* = 2000) with self-reported aphantasia reported more difficulties with face recognition and autobiographical memory relative to 200 participants with vivid visual imagery [102].

Memory problems can be considered a quite broad and non-specific term, and it was deliberately formulated in lay language in this survey study and guided with the description as *‘**remembering things, details, or personal past events’*. This was done with the aim of eliciting responses related to various forms of memory. It is notable in the qualitative data that both Good Memory and Memory Problems were amongst the most frequently used codes. The statements indicate that some participants, but not the majority, described problems related to autobiographic memory: *‘remembering names, holidays, places we have been, events I have been to, films I have seen’*. At the same time, many also reported good semantic memory or good memory for pieces of music, e.g.: *‘I am good at remembering historical facts (people, events, dates) and other random facts; taking multiple-choice tests’.* Thus, the memory components the participants feel skilled at might not be the same ones that cause trouble to others.

*Musicality*. Both good Musical Ability (*n* = 26) and Lack of Musicality (*n* = 17) were reported by participants. For the former, ability to play an instrument and play by ear were often mentioned. For the latter, tone deafness and difficulty singing in tune were often reported, which seems to fit with reports of tone deafness in some cases of DP [103] and impaired pitch discrimination in others [37]. This could also support the argument that face recognition can co-occur with deficits in other modalities.

*Navigation Problems or Abilities*. Navigation problems (*n* = 41) was the most frequently represented category in the qualitative data. Not many reported good navigation abilities. Impaired navigation ability is not an official diagnosis, but is occasionally referred to as (developmental) topographical disorientation [104]. In the current sample, 29 participants with s-DP (25.2%) scored ≤3 on all five the Wayfinding Questionnaire items, which has been suggested to indicate clinically relevant complaints about navigation ability [73].

A few studies have investigated topographical orientation and navigation ability in DP, and there are numerous anecdotal reports of co-occurrence [44]. Existing studies have diverging conclusions: The current observations align with one study which concludes that deficits in navigation, particularly topographical memory, tend to co-occur with DP (*N* = 9) [31]. Another study finds that topographical deficits (cognitive maps formation) are less frequent in DP (*N* = 7) [46]. As opposed to the current study, these previous findings stem from formal testing of different topographical skills implicated in navigation, but the sample sizes are rather small [25]. Interestingly, Bate, Adams, Bennetts, and Line [15] found deficits in retrieval and formation of cognitive maps in a DP case and point out that 6/7 of the sample in the study by Corrow and colleagues [46] did not self-report navigation problems in an initial interview. This of course highlights a recurring issue in DP research (and in general) in that self-report and objective measures do not always coincide.

*Object Agnosia*. The relationship between face and object recognition has perhaps been the most central debate in DP [25], as it concerns the very nature of the architecture of the visual recognition system: whether faces and objects are processed by distinct (domain-specific) or common systems (domain-general) [105]. Twenty-nine participants (25.2%), a fourth of this sample, reported object agnosia as a part of the close-ended items. This, however, was not reflected in the qualitative data. Findings regarding prevalence and severity of deficits in object recognition in DP are mixed, which also applies to whether or not a clear dissociation exists. Regardless, it is clear that DP can occur both with and without self-reported impairments in non-face object recognition [19,20,24,106,107,108,109,110,111,112]. As the current findings are based on self-reported deficits in object recognition, they do not advance the debate further, as studies with formal assessments of face and object recognition (for a review of DP cases up to 2016 see Geskin and Behrman [25]), as well as neuroimaging studies [113,114], still fail to provide consistent results. It does, however, seem safe to conclude that deficits in object recognition are present in at least some cases of DP, as is indicated in this sample as well as in the literature.

*Reading and Spelling*. The participants generally reported more instances of good reading and spelling ability, and few instances of the opposite. The abilities within reading/spelling are many, e.g., flair for languages, literacy, and learning to read from an early age. Few participants reported difficulties with reading or dyslexia on the direct question (3.5% compared with prevalence of childhood dyslexia of 5–10% [115,116,117]), while a few more (7%) reported issues with reading or spelling in the open-ended questions. This aligns with research demonstrating a double dissociation between face recognition and reading abilities [21,22,23,51], although dyslexia and face recognition deficits can also co-occur [32,47,48,49,50]. This suggests that self-reported dyslexia is not more likely to be present with s-DP than in general, nor do the scores on the selected the Adult Reading History Questionnaire items indicate reading difficulties in the sample (Table 3).

*Social and Interpersonal Skills*. Most descriptions of inadequacy in social and interpersonal skills were directly related to face recognition problems, and thus did not receive separate coding in the qualitative analysis. There were, however, some interesting descriptions of social strengths too, e.g.: ‘Knowing what other people feel, and why they feel it. ‘intuitive profiling’ so to say’. Some examples also refer to the ability to read emotions and show empathy. While we know that DP can be accompanied by anxiety and avoidance of social gatherings [3], our results may suggest that further investigations of social coping and explorations of when individuals with DP feel socially capable might prove interesting as well.

*Sports Difficulties/Skills* and *Motor Coordination and Body–Space Perception*. There were a number of reports of Difficulties with Sport (*n* = 17), but also of good Sports Skills (*n* = 12). Sports Skills were often described with an example of excelling in one particular discipline. For Difficulties with Sport, the statements often reflected a broader lack of skill important for many sports. Some statements overlap with experiences coded as Motor Coordination and Body–Space Perception, and are coded with both categories e.g., ‘I have a hard time with things that require hand-eye coordination like playing sports or physical games’. Moreover, besides being clumsy, there are elements of having a hard time judging the distance between oneself and other objects in Motor Coordination and Body–Space Perception. One particularly interesting example is:

‘(…) At work we have to make sure we do not walk within 5 m of an operational machine. I cannot imagine how far that is. I need to memorize the spots where the machines are often positioned, and memorise where I can walk based on where others have walked safely. I will not pass a machine in an unfamiliar spot without instruction from driver or my manager to do so because I can’t tell if I’m at a safe distance.’

As far as we are aware, there is no existing literature on difficulties with motor coordination or body–space perception related to DP, and no investigations of the possible relationship between DP and developmental coordination disorder.

*Skills in Visual Arts*. Some of the participants described talents within primarily drawing and painting. A few described themselves as established artists or/and receivers of awards for their work. For example:

‘I am very artistic. I have shown my work and been featured in a variety of mediums throughout my whole life. Most specifically I paint faces and have since I was about 12 years old and was fascinated with faces.’

To our knowledge, there are no studies regarding the artistic ability of participants with DP, although there are examples in popular media.

*Synesthesia*. Twelve (10.4%) participants with s-DP reported synesthesia. There are few published estimates of synesthesia prevalence in the adult population, and these vary widely, but one well-controlled study reported a prevalence of 1.1–4.4%. [118]. Investigations of synesthesia involve some measure of self-report, but Simner and colleagues’ [118] measures were much more sophisticated than in the present study, potentially limiting the comparability.

The quantitative reports do not align with the qualitative analysis, where only two descriptions of synesthesia are represented. This might be caused by the fact that synesthetic experiences might not instantly be categorized as difficulties or strengths, and thus are not reported in the open-ended questions. Interestingly, it has been shown that some individuals with grapheme-color synesthesia have abnormal face processing measured with, e.g., The Cambridge Face Perception Test (*N* = 3) [57] and CFMT (*N* = 16) [56]. It has been suggested that synesthesia develops in response to complex categorical learning challenges as a perceptual aid [119] or incomplete synaptic pruning during development [120]. Ulimoen and Sørensen [56] hypothesize that in the case of synesthesia, atypical categorization processes are not limited to the inducer stimulus, e.g., a letter inducing color, but might apply to other categories as well, e.g., faces, which could serve as a preliminary interpretation for the current reports.

### 5.2. General Discussion

#### 5.2.1. High Rates of Self-Reported Developmental Comorbidity in s-DP

The amount of developmental comorbidity reported by the sample was quite astonishing, as 65 participants (56.5%) reported suffering from at least one of the following: ADHD, aphantasia, dyscalculia, dyslexia, memory problems, object agnosia, or/and synesthesia. In addition, a large number of participants reported navigational problems on the questionnaire items. The qualitative data revealed that a large number of participants experienced problems in specific domains, in particular with imagery, math, memory, and navigation. However, for all these domains, there were also reports of good abilities—indicating that the link between any of these domains and DP might not be entirely systematic (i.e., these functions might be both associated with and dissociated from DP). To our knowledge, this study is the first to address the possible overlap between DP and other neurodevelopmental disorders and impairments on such a large scale. Our findings seem to suggest a curious link between at least some of the developmental comorbidities [16,28,77]. This could perhaps be interpreted as indicating that DP might be a more disabling disorder than previously thought [15], and/or potentially also less domain-specific and not selective to deficits in facial processing. Another way of interpreting the current findings is that face recognition impairments might be an important node in a developmental co-morbidity space and should be investigated further in other developmental disorders. Even if DP exists in a pure form, the current results suggest that face recognition might be affected or impaired in a range of developmental disorders.

An additional explanation for the high rates of comorbidity observed could be that developmental comorbidity has not been in focus in the literature on DP, and it has not been customary to screen for or ask DP participants about other disorders besides ASD and brain injury [1,11]. Many studies, for instance, do not screen for or exclude participants with dyslexia, and even fewer include observations regarding the other impairments and disorders considered here.

#### 5.2.2. Limitations

While the strength of the present study is in the exploratory and broad approach to the investigation of developmental comorbidity in self-reported DP in a fairly large sample, there are of course also limitations. The main limitation is the lack of a control group without s-DP, which leaves unknown what the frequencies of reports would be in the general population. In addition, all our results are based on self-reports, and while some of our questions are taken from published and validated instruments, we also included yes/no questions regarding the presence of a given condition without the use of validated self-reports or clinical judgement. Thus, our results should be viewed only as a first step in the investigation of possible developmental comorbidity [121,122].

While the PI20 can indicate whether people have trouble with face recognition, it cannot stand alone in determining whether someone is suffering from DP. It is debated within the field how valid self-report instruments like the PI20 are in measuring actual face recognition problems, and relatedly, how much insight people actually have into their facial recognition performance. It has been shown that people in general have only modest insight into their own performance on objective measures of face recognition abilities [9,121,123], but that DPs may have better insight into their objective performances in comparison [121]. Some researchers regard self-reports of face recognition as highly valid due to moderately high correlations (often ~0.35–0.55) between the PI20 (as well as other self-report measures of face recognition) and objective measures of face processing in typical individuals and prosopagnosics [71,124,125,126,127]. In contrast, Arizpe et al. [43] concluded that the PI20 is not sensitive enough to be used as a screening tool for DP. To be clear, the PI20 or other self-report instruments should not be used in isolation for diagnosing DP. Convergent objective testing of face recognition performance in addition to self-report measures is imperative, and on this there is increasing consensus in the field [11,43,127]. Thus, although we based this exploratory study on self-reported face recognition impairment using the PI20, we do not consider this sufficient for an actual diagnosis of DP.

Thus, it is possible that not all participants included in our study would qualify for a diagnosis of DP. Bate and colleagues [60] found that only 38.2% of a sample of 165 self-referred DPs met criteria for impairment on 2/3 tests of face processing. Hence, it seems reasonable to assume that the same could be true for the sample in question. On the other hand, it is also possible that some of the participants have another main disorder with accompanying DP or face recognition deficits, so that the amount of self-reported developmental comorbidity is not due to DP but other primary neurodevelopmental disorders [28] or developmental neural vulnerability [16,77,128]. There is also an important caveat in interpreting the frequencies with which specific disorders or deficits were reported, as we did pre-specify some of these in the questionnaire. Thus, had we included other specific disorders, e.g., developmental coordination disorder or OCD, our findings would likely look different. The mentioning of specific disorders in the questionnaire could also prime the participants to consider impairments they would not have considered otherwise, e.g., aphantasia, which was mentioned specifically and was frequently reported both in the quantitative and qualitative data. This, however, does not detract from the main finding that a large proportion of the included s-DP reported indications of additional impairments and disorders.

Relatedly, recruiting from online fora might bias the sample towards people with particular interest in their suspected condition or in the research question, and perhaps also people with an interest in and experience of other diagnoses. We are not aware of any studies on bias in online recruitment in relation to prosopagnosia, but based on the general literature on self-selection and online recruitment [129,130] this is an important caveat to bear in mind.

In addition, the current sample size of 115, and the additional metrics from the exploratory factor analysis, lies in the lower end to satisfy the criteria of Maccallum et al. [131]. There are, however, differing views on the right sample size for this large sample procedure [132].

For the qualitative data, the aim of applying the open-ended questions was to elicit more detailed descriptions that expanded our understanding of the phenomena under investigation, however, not all responses were detailed. For example, some participants would describe their difficulties or abilities by listing them very briefly like this: *‘Memory, navigation, math’.* As this was an online survey with unnamed participants, there was also no way we could ask clarifying questions in relation to statements that were arbitrary or lacked detail.

## 6. Conclusions

The observed high co-occurrence of developmental comorbidity in self-reported DP may have important implications for our understanding of neurodevelopmental disorders and conditions and their interplay. Particularly, high rates of self-reported synesthesia, navigational difficulties, aphantasia, and memory problems (particularly autobiographical memory) and diversity within subjective difficulties and strengths, e.g., math skills and motor coordination, suggests that there is ample room for future research on this topic. Investigating the relationship of DP to other developmental conditions in more detail, and whether face recognition deficits may exist in other neurodevelopmental disorders, might prove fruitful in characterizing the neurodevelopmental co-morbidity space or overlap between disorders more fully. While we have taken diagnostic categories and pre-defined cognitive domains as our starting point here, further work might go beyond the ‘core-deficit’ approach [133], and our findings suggest that face recognition may be an important aspect to assess when aiming to map transdiagnostic features of developmental disorders [133,134] in the future.

## Figures and Tables

**Figure 1 brainsci-12-00230-f001:**
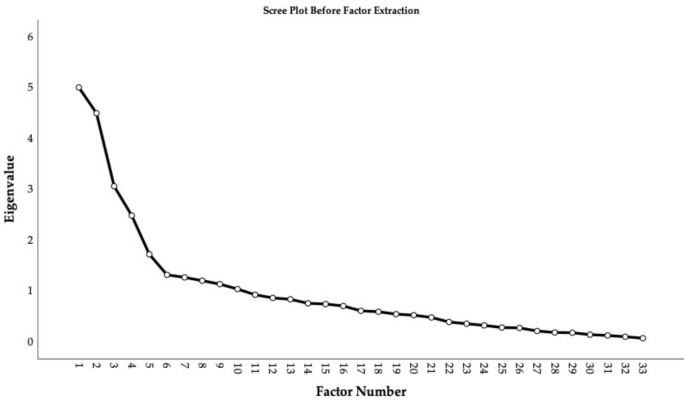
Scree plot before factor extraction.

**Table 1 brainsci-12-00230-t001:** Participants’ age.

Age in Years	*n (%)*
18–24	2 *(1.7)*
25–34	18 *(15.7)*
35–44	38 *(33.0)*
45–54	23 *(20.0)*
55–64	18 *(15.7)*
65–74	14 *(12.2)*
≥75	2 *(1.7)*
Total	115 *(100.0)*

**Table 2 brainsci-12-00230-t002:** Demographics.

Demographics	*n (%)*
DP diagnosis *	34 *(29.6)*
English as first language	77 *(67.0)*
Handedness (left/right/ambidextrous)	10/99/6 *(8.7/86.1/5.2)*

* Self-report of being diagnosed with DP by healthcare professional or researcher.

**Table 3 brainsci-12-00230-t003:** Self-reported developmental comorbidity co-occurring with s-DP.

Self-Reported Developmental Comorbidity	*n (%)*
Presence of at least one developmental comorbidity ^a^	65 *(56.5)*
Dyslexia	4 *(3.5)*
Dyscalculia	4 *(3.5)*
ADHD	7 *(6.1)*
Aphantasia	24 *(20.9)*
Synesthesia	12 *(10.4)*
Object agnosia	29 *(25.2)*
Memory problems	27 *(23.5)*
Presence of not-predefined mental, cognitive or intellectual disorder ^b^	13 *(11.3)*
Presence of not-predefined conditions related to vision ^b^	2 *(1.7)*
Presence of not-predefined somatic disorder ^b^	2 *(1.7)*
Navigation problems *	29 *(25.2)*

Note that for the individual disorders, a participant reporting, e.g., both ADHD and dyslexia, is counted in both respective categories. ^a^ Number of participants reporting at least one comorbid disorder or deficit being either dyslexia, dyscalculia, ADHD, aphantasia, synesthesia, object agnosia, or/and memory problems. Navigation problems are not counted here as they are not a binary variable. ^b^ Details about the specific self-reported conditions can be found in Appendix A. Not-predefined refers to conditions reported in the free-text options. * Navigation problems are reported as the number of participants reporting indications of clinically relevant complaints (≤3) on all five items from the Wayfinding Questionnaire.

**Table 4 brainsci-12-00230-t004:** Number of reported developmental comorbidities (besides s-DP).

Number of Developmental Comorbidities *	*n (%)*
0	50 *(43.5)*
1	34 *(29.6)*
2	22 *(19.1)*
3	8 *(7.0)*
5	1 *(0.9)*
Total	115 *(100.0)*

* These percentages are based on participants reporting as having ADHD, dyslexia, or dyscalculia to the questions ‘Do you have other diagnoses? Particularly related to vision, mental illness, learning difficulties, sensory disorders, developmental disorders, motor disorders, etc.’ or aphantasia, synesthesia, object agnosia, and/or memory problems in response to the question ‘Do you have any of the following conditions related to cognitive strengths or difficulties?’. Thus, navigation problems are not included.

**Table 5 brainsci-12-00230-t005:** Developmental comorbidity and s-DP severity.

Severity Group Based on PI20	Number of Cases Reporting Developmental Comorbidity (% of Subsample)
Mild DP (*n* = 15)	9 *(60.0)*
Moderate DP (*n* = 42)	22 *(52.4)*
Severe DP (*n* = 58)	34 *(58.6)*

**Table 6 brainsci-12-00230-t006:** Factor loadings based on a principal axis factoring with oblique rotation for 33 items from the Difficulties and Abilities in Developmental Prosopagnosia Questionnaire (*N* = 115).

Difficulties and Abilities in Developmental Prosopagnosia Questionnaire Items	Factor 1	Factor 2	Factor 3	Factor 4
PI1. My face recognition is worse than most people	0.456		−0.173	−0.165
PI2. I have always had a bad memory for faces	0.289	0.165		
PI3. I find it noticeably easier to recognize people who have distinctive facial features			−00.331	
PI4. I often mistake people I have met before for strangers	0.430			
PI5. When I was at school I struggled to recognize my classmates	**0.653**		0.173	0.106
PI6. When people change their hairstyle, or wear hats, I have problems recognizing them	0.478			−0.196
PI7. I sometimes have to warn new people I meet that I am ‘bad with faces’	0.411			
PI8. I find it easy to picture individual faces in my mind	0.154			−0.160
PI9. I am better than most people at putting a ‘name to a face’	0.152			
PI10. Without hearing people’s voices, I struggle to recognize them	**0.548**		0.115	
PI11. Anxiety about face recognition has led me to avoid certain social or professional situations	0.377			0.183
PI12. I have to try harder than other people to memorize faces		0.141	−0.330	
PI13. I am very confident in my ability to recognize myself in photographs	0450	0.171		
PI14. I sometimes find movies hard to follow because of difficulties recognizing characters	0.371		0.209	−0.166
PI15. My friends and family think I have bad face recognition or bad face memory	0.472	−0.141		
PI16. I feel like I frequently offend by not recognizing who they are	**0.549**	−0.141		
PI17. It is easy for me to recognize individuals in situations that require people to wear similar clothes (e.g.,suits, uniforms, swimwear)	0.464		−0.315	−0.102
PI18. At family gatherings, I sometimes confuse individual family members	**0.553**			
PI19. I find it easy to recognize celebrities in ‘before-they-were-famous’ photos, even if they have changed considerably	0.149		−0.182	−0.326
PI20. It is hard to recognize familiar people when I meet them out of context (e.g.,meeting a work colleague unexpectedly while shopping)	**0.646**			0.199
WQ1. I am good at understanding and following route descriptions		**−0.801**		−0.124
WQ2. I can always orient myself quickly and correctly when I am in an unknown environment		**−0.967**		
WQ3. I can easily find the shortest route to a known destination		**−0.911**		
WQ4. I can usually recall a new route after I have walked it once		**−0.853**	−0.102	
WQ5. When I am in a building for the first time, I can easily point to the entrance of this building.		**−0.862**		
ARHQ1. How much extra help did you need when learning to read in elementary school?			**0.767**	
ARHQ2. How would you compare your reading skill to that of others in your elementary classes?	−0.140		**0.744**	
ARHQ3. How much difficulty did you have learning to spell in elementary school?	0.166		**0.695**	0.159
ARHQ4. How much difficulty did you have learning to read in elementary school?			**0.829**	
ARHQ5. How would you compare your current reading speed to that of others of the same age and education?		0.171	**0.558**	−0.120
MATH1. As a child, did you make careless errors in math, such as adding when the sign indicated subtraction?	0.156		0.108	**0.779**
MATH2. As a child, did you have trouble learning new math concepts?		0.128		**0.816**
MATH3. Do you have difficulties estimating quantities, e.g., how many coins are lying on a table?		0.194		**0.653**

Note: Values below 0.10 are suppressed. PI = prosopagnosia index, WQ = Wayfinding Questionnaire, ARHQ = Adult Reading History Questionnaire. **Bold type**: Strong factor loadings of above 0.5 and below −0.5. Factor loadings not meeting the criterion of above 0.32 or below −0.32 are written in grey.

**Table 7 brainsci-12-00230-t007:** Qualitative coding categories (*n* = 114) ^a^.

Difficulties Codes	*N* ^b^	Quote	Strengths Codes	*N* ^b^	Quote
Aphantasia	24	*‘I am not very good at visualizing things in my head. I’ve explained that it’s more like when I picture something in my mind, it looks kind of like a Lite Brite toy. With just like colored outlines of things. So imagining The beach, it’s just a yellow line for the sand, some blue blur for the ocean, and an outlined yellow sun. Like a kid drawing.’*	Good Visual Mental Imagery	7	*‘I have a very strong visual imagination. I can create whole worlds full of detail in my imagination.’*
Difficulties with Math	27	*‘I think I have some dyscalculia. I will transpose numbers even though I am trying to be careful and can do this even in a short string of only 3 numbers. However I don’t have trouble with mathematical concepts or at least not up to the usual school leaver level.’*	Mathematical Abilities	20	*‘I am better at math than most people. I enjoy math and I’m good at it. Even when I don’t “feel” like I fully understand it, I perform well on math/stats tests.’*
Reading and Spelling Difficulties	8	*‘Learning a new language is impossible (I live in Indonesia for 23 years). Very slow reader, I easily forget what I have read.’*	Reading and Spelling Abilities	39	*‘I’m a very good reader, writer, editor, and proofreader.’*
Memory Problems	21	*‘I have a terrible memory. I feel like I barely remember anything prior to highschool and I am only 29 years old. Even then things are very foggy. I only remember things if I see them in a photograph.’*	Good Memory	40	*‘Far above average skilled in remembering fun facts and historical events. Very good memory in order of recalling details of what’s happened in my own life.’*
Object Agnosia	7	*‘I’m ‘car-blind’, and had to memorize my parents’ license plates as a child.’*			
			Synesthesia	2	*‘See colours when listening to music’*
Lack of Musicality	17	*‘Very tone deaf- measures by up to 2 full notes by a music teacher.’*	Musicality	26	*‘I can hear something once and play it on the piano.’*
Difficulties with Sports	17	*‘I am a disaster when it comes to sports or athletics, especially team sports. Partially this is due to my lack of coordination and skill, and partially because I never knew who to pass the ball to!’*	Sports Skills	12	*‘I was a highly competitive gymnast for 14 years. I have very good body awareness.’*
Motor Coordination and Body–Space Perception	10	*‘I’m clumsy, can’t catch or throw, and miss judge and catch the edges of things when walking.’*			
Problems with Inattentiveness, Impulsivity, or Hyperactivity	2	*‘Need more time on tasks, partly because I want to be thorough and am afraid of doing mistakes, partly because of lack of routines. Bad at routines and “what to do next”. Easy to distract. “Colour codes” on buttons* etc. *doesn’t help me, seems like I don’t notice them (I have normal colour vision).’*			
			Social and Interpersonal Skills	12	*‘Knowing what other people feel, and why they feel it. “intuitive profiling” so to say’*
			Skills in Visual Arts	12	*‘I am very artistic. I have shown my work and been featured in a variety of mediums throughout my whole life. Most specifically I paint faces and have since I was about 12 years old and was fascinated with faces.’*
Navigation Problems	41	*‘I am terrible at navigation. Even in a mid-sized town (100,000 people) I have lived for 20 years, I still get lost. I have no clue which is N, S, E, W.’*	Navigation Abilities	13	*‘Maps, geography.’*

Full descriptions of the coding categories are available in the coding manual in the Appendix A. ^a^ One participant wrote the response in French and was thus excluded from the qualitative analysis. ^b^ The frequency is based on the number of participants mentioning content related to the specific code; even if the same person has several codes for the same category, it is only counted once. The background indicates separate interpretations/codings; Italics indicate quotations.

## Data Availability

The data presented in this study are openly available in an OSF-storage at https://osf.io/752pk/ (accessed on 14 December 2021).

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
