# Peer review of "Is It Just Face Blindness? Exploring Developmental Comorbidity in Individuals with Self-Reported Developmental Prosopagnosia"

_brainsci, 2022, doi:10.3390/brainsci12020230_

Round 1
Reviewer 1 Report
This study investigates the comorbidity of developmental prosopagnosia with a range of cognitive abilities such as reading and navigation. To my knowledge, this is the largest and most in-depth study of this phenomena. The study includes both quantitative and qualitative data, and finds that 57% of DP’s (as defined by self-reported questionnaire) have comorbid issues. Comorbidity of prosopagnosia with other cognitive abilities (in particular, visual processing) is a hot topic, so this is a timely study, which I’m sure will contribute to the field.
MAJOR COMMENTS
- The main issue with this study is the lack of independent diagnosis of DP in the participants; however, the authors discuss this issue in the discussion and note the limitations.
MINOR COMMENTS
Abstract:
- Good abstract – I have no comments or suggestions.
Introduction:
- Good introduction – I have no comments or suggestions.
Material and Methods:
- Section 2.4.2 – missing what I’m assuming is an alpha level? “(? = 0.96)”.
- Section 2.4.2 – where did you get the correlations and p values (and alpha levels) from? Did your participants also complete the CFMT? If so, this needs to be made clear in your methods section. If a previously published paper, then please cite.
- Section 2.4.4 – missing what I’m assuming is an alpha level again? “(? = .94 and 0.92)”.
Data Analysis:
- I have no comments or suggestions.
Results:
- I have no comments or suggestions.
Discussion:
- Line 494 - “Given the suggested link between aphantasia and SDAM” – please specify what SDAM is.
Author Response
Response to Reviewer 1:
This study investigates the comorbidity of developmental prosopagnosia with a range of cognitive abilities such as reading and navigation. To my knowledge, this is the largest and most in-depth study of this phenomena. The study includes both quantitative and qualitative data, and finds that 57% of DP’s (as defined by self-reported questionnaire) have comorbid issues. Comorbidity of prosopagnosia with other cognitive abilities (in particular, visual processing) is a hot topic, so this is a timely study, which I’m sure will contribute to the field.
Response: We thank Reviewer 1 for the positive assessment of our study and manuscript.
MAJOR COMMENTS
- The main issue with this study is the lack of independent diagnosis of DP in the participants; however, the authors discuss this issue in the discussion and note the limitations.
MINOR COMMENTS
Abstract:
- Good abstract – I have no comments or suggestions.
Introduction:
- Good introduction – I have no comments or suggestions.
Material and Methods:
- Section 2.4.2 – missing what I’m assuming is an alpha level? “(? = 0.96)”.
Response: Thanks for pointing out that this was unclear. What we report here is Chronbach’s a, which we now specify in the text.
- Section 2.4.2 – where did you get the correlations and p values (and alpha levels) from? Did your participants also complete the CFMT? If so, this needs to be made clear in your methods section. If a previously published paper, then please cite.
Reponse: All metrics in this paragraph are taken from the original validation study of the PI20 (Shah et al., 2015). We have now modified the text as follows, to make this more clear:
“The psychometric properties of the PI20 are good: in the original validation study by Shah et al. [70] it was shown to have very high internal consistency (Chronbach’s a = 0.96), and good convergent validity as it correlates well with objective measures of face recognition, e.g., CFMT (r=-.68, p<.0001). The maximum score is 100, and higher scores indicate more difficulty recognizing faces. Scores of 65-74, 75-84, 85-100 can broadly indicate mild, moderate and severe DP, respectively [70].”
- Section 2.4.4 – missing what I’m assuming is an alpha level again? “(? = .94 and 0.92)”.
Response: Theses values also referred to Chronbach’s alfa, which is now specified in the text.
Discussion:
- Line 494 - “Given the suggested link between aphantasia and SDAM” – please specify what SDAM is.
Response: As requested by reviewer 3, we have limited the use of abbreviations, and now write severely deficient autobiographical memory rather than SDAM in this sentence.
Reviewer 2 Report
The current study is an interesting, timely, and potentially informative (though see below) investigation of the self-reported comorbidities of individuals with self-reported face recognition difficulties. I found that the manuscript was well-written, appreciated the application of factor analyses, and thought that there were several potential insights to add to the ongoing prosopagnosia debates over face specificity and presence of potential object/word/reading deficits.
Though I found the manuscript to have several strengths, I have a couple of serious issues. First, there is the major concern with classifying participants as developmental prosopagnosics (DP) when only using self-reports and not having any evidence from objective face recognition tests. Though a few DP papers have been published using this self-report-only approach (e.g., Kennerknecht et al., 2006), not having objective tests when diagnosing DP has received quite a bit of criticism (e.g., Duchaine et al., 2008; Arizpe et al., 2019) and to my knowledge, all recent DP studies have required evidence of objective face recognition deficits. Notably, the correlation between self-reported and objective face recognition performance only ranges from r=.2-.4, suggesting that there are many situations where self-reported face recognition deficits do not result in a DP diagnosis. Perhaps the authors could reframe the paper as 'those with self-reported face recognition difficulties' instead of DP, but I fear that this wouldn't map onto existing literatures very well.
A second issue is that there is no comparison of the prevalence of comorbidities in the self-reported DP group with a control group. This makes interpretation of the comorbidity results quite difficult because individuals who join health-related or neuro-based support groups may have slightly elevated comorbidities to begin with. It would also be important to collect gender data from participants (or even do so now if you can), as men and women have different risks for comorbidities at different ages. Finally, I did not find the justification of why autism traits were not measured to be very convincing. This is a very common DP comorbidity and there are validated self-reports to characterize ASD (autism quotient questionnaire).
Author Response
Response to Reviewer 2:
The current study is an interesting, timely, and potentially informative (though see below) investigation of the self-reported comorbidities of individuals with self-reported face recognition difficulties. I found that the manuscript was well-written, appreciated the application of factor analyses, and thought that there were several potential insights to add to the ongoing prosopagnosia debates over face specificity and presence of potential object/word/reading deficits.
Response: Thanks for this nice evaluation, and for a thoughtful reading of our manuscript.
Though I found the manuscript to have several strengths, I have a couple of serious issues.
- First, there is the major concern with classifying participants as developmental prosopagnosics (DP) when only using self-reports and not having any evidence from objective face recognition tests. Though a few DP papers have been published using this self-report-only approach (e.g., Kennerknecht et al., 2006), not having objective tests when diagnosing DP has received quite a bit of criticism (e.g., Duchaine et al., 2008; Arizpe et al., 2019) and to my knowledge, all recent DP studies have required evidence of objective face recognition deficits. Notably, the correlation between self-reported and objective face recognition performance only ranges from r=.2-.4, suggesting that there are many situations where self-reported face recognition deficits do not result in a DP diagnosis. Perhaps the authors could reframe the paper as 'those with self-reported face recognition difficulties' instead of DP, but I fear that this wouldn't map onto existing literatures very well.
Response: This a central issue regarding out paper, and one we have aimed to take seriously. This is why we mention in the title, that we have studied “Individuals with self reported developmental prosopagnosia”, and refer to the recruited sample in the text as having ‘self reported developmental prosopagnosia’ – abbreviated to DPs. Based on this comment and one by reviewer 3 – we now also discuss this further in the paper, and refer to the study by Arizpe et al., 2019 which addresses the relationship between self report and objective testing of face recognition performance in in section 5.2.2. Limitations. This paragraph now reads:
“Some researchers regard self-reports of face recognition as highly valid due to moderate-high correlations (often ~.35-.55) between the PI20 (as well as other self-report measures of face recognition) and objective measures of face processing in typical individuals [70,122-124]. In contrast, Arizpe, et al. [125] conclude that the PI20 is not sensitive enough to be used as a screening tool alone, as done in the present study. Along the same vein , it has been shown that people in general have only modest insight into their own performance on objective measures of face recognition abilities [9,120,126], but that DPs have better insight into their objective performances in comparison [120].
Thus, it is possible that not all participants included in our study would qualify for a diagnosis of DP. Bate and colleagues [59] found that only 38.2 % of a sample of 165 self-referred DPs met criteria for impairment on 2/3 tests of face processing. Hence, it seems reasonable to assume that the same could be true for the sample in question.”
- A second issue is that there is no comparison of the prevalence of comorbidities in the self-reported DP group with a control group. This makes interpretation of the comorbidity results quite difficult because individuals who join health-related or neuro-based support groups may have slightly elevated comorbidities to begin with. It would also be important to collect gender data from participants (or even do so now if you can), as men and women have different risks for comorbidities at different ages.
Response: It is correct that it would have been helpful with a control sample to compare our results too, but such data were not collected during this exploratory project. This is why we have compared with published prevalence rates where this was deemed appropriate. Given a possible bias in online recruitment from interest-groups, we now address the possibility that people who join the online groups we recruited from may not be representative of the DP-population as a whole (section 5.2.2. Limitations):
“Relatedly, recruiting from online fora might bias the sample towards people with particular interest in their suspected condition or in the research question, and perhaps also people with an interest in and experience of other diagnoses. We are not aware of any studies on bias in online recruitment in relation to prosopagnosia, but based on the general literature on self-selection and online recruitment [128,129] this is an important caveat to bear in mind.”
See also response to Reviewer 3’s point 1.
Regarding gender data, see also our response to Reviewer 3’s point 16. We agree that this issue is important and that such data could be informative. However, for this explorative study in which participants remained anonymous we were advised by our data-protection officer to refrain from gathering data on gender to preserve anonymity. Because the participants are anonymous to us, we cannot go back and collect these data now, but this issue will be important to address in future research.
- Finally, I did not find the justification of why autism traits were not measured to be very convincing. This is a very common DP comorbidity and there are validated self-reports to characterize ASD (autism quotient questionnaire).
Response: This is a good point, and something we considered carefully before starting the project. In retrospect, including the ASQ or similar instead of excluding participants with possible ASD would have been really interesting. The choice to exclude participants with ASD was made because most studies of DP exclude participants with ASD, and because we were worried that particpants with co-occuring ASD and DP could be driving the results regarding the occurrence of other neurodevelopmental disorders, obscuring the link between DP and other comorbidites. We have now tried to clarify our reasoning in the introduction:
“The exception is the relation to autism spectrum disorder (ASD), where observations of face recognition deficits are common [33-35]. However, ASD often presents with severe social and intellectual deficits not found in DP, which is why ASD is one of the most common exclusion criteria from DP studies [1]. In addition, ASD can co-occur with a range of other neurodevelopmental disorders independently of face processing deficits. For these reasons the potential overlap between ASD and DP is not considered in the present study, and participants with ASD were excluded from participation.”
Reviewer 3 Report
The present research examined the potential co-morbidities present in developmental prosopagnosia. This was an exploratory analysis as this is the first study of its kind. For this reason, the authors did not lead with any particular hypothesis. They tested 115 individuals with self-reported developmental prosopagnosia using an online survey asking about face recognition skills, as well as a list of potential comorbid developmental conditions, largely selected based on prior literature. In addition, they asked open-ended questions about the strengths and weaknesses of the participants. Overall, 57% of participants reported at least one comorbidity, however, the impairments reported were highly diverse with some participants listing a skill as a strength, and others listing the same skill as a weakness.
I am really happy to see that this work is being done as I feel it is a long-overlooked domain of research within the sub-field of prosopagnosia. While the study suffers from the limitations one would expect for an exploratory study of this kind, this study provides the necessary contribution to begin conversations and further research on the study of comorbidities in developmental prosopagnosia. In addition, the paper is well written.
Below, I include suggestions for improving the paper. When appropriate, I include page and line number (pg; line).
- The main limitation of this study is the lack of non-prosopagnosic control group. It makes the finding of “57%”, and other prevalence rates included, rather uninterpretable. While 57% seems quite large, it is not unreasonable to expect that a similar proportion of people without prosopagnosia would report difficulties in at least one of the domains tested, especially considering that they didn’t have to report an official diagnosis in order to be included in that group. The primary fix for this would be to collect online data from an equal number of non-prosopagnosic controls! If not, at the very least, I would step away from the focus on the overall prevalence of comorbidity (or the prevalence of any particular condition tested) and instead focus on relative comparisons between the different comorbidities.
- As an addendum to my point above, I do think it is reasonable in a study like this to compare prevalence of a particular comorbidity to prevalence rates reported in the literature, but only in situations in which diagnostic procedure used in the present study is identical or very similar to the diagnostic procedure used to determine the prevalence in prior literature. For example, it seems reasonable to compare the percentage of DP’s who reported an official diagnosis of ADHD to the prevalence of ADHD reported in the general population on the assumption that ways they would be diagnosed would be similar. However, it seems less reasonable to compare the prevalence rate of aphantasia to prior literature, unless prior literature is also based strictly on self-reported data, as in the present study.
- While I understand the standard in the field to exclude those with ASD and other disorders that can impair social functioning, I have two concerns about this: 1) it is fairly well established that developmental disorders tend to co-occur. Therefore, by excluding these individuals, you may actually be underestimating the degree of comorbidity in these cases. 2) When the interest is comorbidities, it seems important to not exclude cases based on comorbidities. However, I realize this is a challenge for the field in a general sense, and not specific to this paper. How can we study comorbidities when we are excluding subjects based on comorbidities?
- Another point that is not specific to this paper, but I think should be addressed is the challenge that the degree to which some of these disordered (e.g. ADHD) are diagnosed has changed over time and with advances in our understanding of developmental disability. I am curious if this could be addressed in some way in this paper, especially given that your subjects span from 18-75+ years of age.
- While I understand that, as an exploratory analysis you didn’t have specific a priori However, especially for the factor analysis, it would be useful to specify at the end of the introduction, different ways the data could come out and how that would be interpreted. For example, “one possibility is that the factor analysis produces four factors. This would suggest that . . . “ Or something like that. What if it came out with more than 4? What about 3? What percentage of people would have to report comorbidities for you to find it meaningful? Why? In general, in lieu of an explicit hypothesis, it would be great to consider possible outcomes of the data and how those outcomes might be interpreted.
Minor Points
Intro
- (1/37): I get nervous when I see 2-2.5% reported as the prevalence rate of developmental prosopagnosia on the grounds that many researchers use 2 SD below the mean as a diagnostic cut-off. Can you include at least a mention of this or a reference to a study that points it out?
- (2/69) The authors argue that “previous studies have not explicitly addressed potential co-occurring ND deficits. However, on line 54 of the same page they cite studies have examined the co-occurrence of DP with topographic disorientation and dyslexia.
- The extensive use of acronyms makes the study a bit hard to follow. I suggest reducing the number of acronyms used to just those that are well known (e.g. ADHD).
Method
- (3/120) Were subjects only included if they reported a lifelong impairment in face recognition?
- (4/146-157) Is it possible that some of the participants did not select “yes” for some of the disorders they were asked about because they don’t know what they are? Was any further clarification provided as to what constitutes dyslexia, for example?
- (4/167-171) Arizpe et al (2019) makes a strong case for not using the PI20 alone as a screening tool for prosopagnosia. I think the limitation of the PI20 being used in this way should at least be mentioned: https://link.springer.com/article/10.3758/s13428-018-01195-w.
- I am happy to see that the authors report the validity metrics of the scales from which they used questions. However, I am not sure that the validity metrics for the entire scale, as reported on pages 4 & 5, can be used to justify the use of individual items from those scales.
- Section 3.1.2: It would be useful if the authors added some evidence that the sample size they used is appropriate for the factor analysis.
- (6/284-286) When you refer to “a meaningful amount of description” are you referring to the individual level or the group level?
- For determining those who have navigational problems, how did you decide to make the cutoff <=3?
- From the standpoint of making sure that there is representation of women in research studies, I find it concerning that biological sex or gender was not recorded.
Results
- Section 4.4 could benefit from some sort of summary of the contents of table 6 and 7, which could then reference the table.
Discussion
- (15/423) What is “the counterpart” referring to?
- (16/470) I do not understand what this sentence is trying to say!
- (16/490) You mention that aphantasia was reported frequently in both the qualitative and quantitative data. Could this be because participants were primed to consider aphantasia when completing the quantitative portion of the study?
- (17/530) N=9/? And N=7/?
- (17/551-552) Object agnosia and “deficits in object recognition” seem, to me, to be different, with object agnosia being much more severe. Therefore, it seems fair to say that DP can occur both with and without impairments in object recognition (lines 545-546] but not that “object agnosia will present with some cases of DP.” I would argue that object agnosia, as an official diagnoses, would exclude a diagnosis of prosopagnosia (if you can’t identify basic level objects, then failure to recognize faces is likely secondary to that).
- When relevant, and considering point #2 above, it is helpful to include the currently reported prevalence rates for each of the disorders examined in the discussion (as currently is the case for many, but not all). For example, do you know what percentage of the general population reports having at least one developmental disability?
Author Response
Response to Reviewer 3:
The present research examined the potential co-morbidities present in developmental prosopagnosia. This was an exploratory analysis as this is the first study of its kind. For this reason, the authors did not lead with any particular hypothesis. They tested 115 individuals with self-reported developmental prosopagnosia using an online survey asking about face recognition skills, as well as a list of potential comorbid developmental conditions, largely selected based on prior literature. In addition, they asked open-ended questions about the strengths and weaknesses of the participants. Overall, 57% of participants reported at least one comorbidity, however, the impairments reported were highly diverse with some participants listing a skill as a strength, and others listing the same skill as a weakness.
I am really happy to see that this work is being done as I feel it is a long-overlooked domain of research within the sub-field of prosopagnosia. While the study suffers from the limitations one would expect for an exploratory study of this kind, this study provides the necessary contribution to begin conversations and further research on the study of comorbidities in developmental prosopagnosia. In addition, the paper is well written.
Response: We thank Reviewer 3 for these positive comments, and are particularly pleased that the reviewer sees our study as the beginning of a conversation about and future research into comorbidity in developmental prosopagnosia, which we too think is very important. We also thank reviewer 3 for the thorough reading of our manuscript and the helpful suggestions for clarifications and improvement.
Below, I include suggestions for improving the paper. When appropriate, I include page and line number (pg; line).
- The main limitation of this study is the lack of non-prosopagnosic control group. It makes the finding of “57%”, and other prevalence rates included, rather uninterpretable. While 57% seems quite large, it is not unreasonable to expect that a similar proportion of people without prosopagnosia would report difficulties in at least one of the domains tested, especially considering that they didn’t have to report an official diagnosis in order to be included in that group. The primary fix for this would be to collect online data from an equal number of non-prosopagnosic controls! If not, at the very least, I would step away from the focus on the overall prevalence of comorbidity (or the prevalence of any particular condition tested) and instead focus on relative comparisons between the different comorbidities.
- As an addendum to my point above, I do think it is reasonable in a study like this to compare prevalence of a particular comorbidity to prevalence rates reported in the literature, but only in situations in which diagnostic procedure used in the present study is identical or very similar to the diagnostic procedure used to determine the prevalence in prior literature. For example, it seems reasonable to compare the percentage of DP’s who reported an official diagnosis of ADHD to the prevalence of ADHD reported in the general population on the assumption that ways they would be diagnosed would be similar. However, it seems less reasonable to compare the prevalence rate of aphantasia to prior literature, unless prior literature is also based strictly on self-reported data, as in the present study.
Response: Regarding the lack of a control group, this is an issue were are very aware of, which is a result a decision that was made early on in this exploratory project. We mention this in the section 5.2.2. Limitations:
“The main limitation is the lack of a control group without DPs, which leaves it unknown what the frequencies of report would be in the general population. In addition, all our results are based on self-reports, and while some of our questions are taken from published and validated instruments, we also included yes/no questions regarding the presence of a given condition without the use of validated self-reports or clinical judgement. Thus, our results should be viewed only as a first step in the investigation of possible developmental comorbidity [122,123].”
The point about limiting comparison of prevalence to situations where the diagnostic procedure is similar to the present project is well taken, we now comment on how prevalence has been estimated directly in the text regarding the different co-morbidities, and have removed reference to published prevalence estimates where diagnostic procedures were clearly too different.
- While I understand the standard in the field to exclude those with ASD and other disorders that can impair social functioning, I have two concerns about this: 1) it is fairly well established that developmental disorders tend to co-occur. Therefore, by excluding these individuals, you may actually be underestimating the degree of comorbidity in these cases. 2) When the interest is comorbidities, it seems important to not exclude cases based on comorbidities. However, I realize this is a challenge for the field in a general sense, and not specific to this paper. How can we study comorbidities when we are excluding subjects based on comorbidities?
Reponse: We agree, and we pondered this quite a bit when designing the study. Our main reasons for excluding participants with ASD were a) that this is common in studies of DP, and b) that we were worried that ASD-comorbidities, rather than comorbidities related to DP would be driving the results. We have now tried to clarify this in the introduction. See also response to Reviewer 1’s point 3.
- Another point that is not specific to this paper, but I think should be addressed is the challenge that the degree to which some of these disordered (e.g. ADHD) are diagnosed has changed over time and with advances in our understanding of developmental disability. I am curious if this could be addressed in some way in this paper, especially given that your subjects span from 18-75+ years of age.
Response: This is an interesting point. Primed by this question we had another look at the dataset, and it does seem that more younger participants report ADHD. Of the 7 participants who reported being diagnosed with ADHD; 4 were 25-34 (n= 18 in this age group), 2 were 35-44 (n=38), and one 64-75 (n=14). However, given the relatively small number that reported having a formal diagnosis at all, and because we recruited on social media which may bias the age groups we recruited from, we do not think our data can elucidate this question in any interesting way.
- While I understand that, as an exploratory analysis you didn’t have specific a priori However, especially for the factor analysis, it would be useful to specify at the end of the introduction, different ways the data could come out and how that would be interpreted. For example, “one possibility is that the factor analysis produces four factors. This would suggest that . . . “ Or something like that. What if it came out with more than 4? What about 3? What percentage of people would have to report comorbidities for you to find it meaningful? Why? In general, in lieu of an explicit hypothesis, it would be great to consider possible outcomes of the data and how those outcomes might be interpreted.
Response: We have considered this suggestion, and though we find it appealing to be able to describe the various ways in which we imagined the data could come out, we also do want to keep the exploratory format and not comment on specific patterns of results in the introduction. Instead, we now write in the introduction:
“Because of the lack of previous knowledge to guide hypotheses, we chose a coarse-grained and exploratory approach, and thus there were no formal statistical hypotheses to be tested. We were, however, particularly interested in the relationship between the severity of self-reported face recognition deficits and presence of comorbidities, and exploring the relationship between ratings of face recognition on one hand, and items regarding navigation, reading, spelling and mathematical abilities. In addition, we were interested to see whether open-ended questions could provide indications of possible comorbidities that have not previously been explored in relation to DP.”
To further address this point, we have also amended section 3.1.2. so it now reads:
“A reasonable assumption could be that a four factor structure would emerge. It has been demonstrated that PI20 has one factor [70], and the Wayfinding Questionnaire , the Adult Reading History Questionnaire and the math items would be expected to represent different abilities, and hence different factors. However, the comorbidity patterns described above [31,32,51], e.g. co-occurring developmental reading and mathematical difficulties, could allow for speculations of correlation between these measures, potentially resulting in three factors. However, in both cases, the knowledge is still too limited to allow for precise predictions of the factor outcome, and thus an exploratory approach was taken.”
Minor Points
Intro
- (1/37): I get nervous when I see 2-2.5% reported as the prevalence rate of developmental prosopagnosia on the grounds that many researchers use 2 SD below the mean as a diagnostic cut-off. Can you include at least a mention of this or a reference to a study that points it out?
Response: This is indeed a contentious issue, and the published prevalence studies do need to be replicated or, preferably, better prevalence studies should be conducted. If the definition of any given disorder was 2 SD below the norm on one given test, we could not separate the tail of the normal distribution from a ‘deficit’ or ‘disorder’. As interesting and impoartant as we find this debate, we settled for the second suggestion – including a reference that points this out, so the sentence now reads:
“This is intriguing, as estimates suggest that 2-2.5 % of the population suffer from this DP [9,10, but see 11 for a discussion]” where [11] is a reference to Barton & Corrow (2016) The problem of being bad at faces, that addresses this issue.
- (2/69) The authors argue that “previous studies have not explicitly addressed potential co-occurring ND deficits. However, on line 54 of the same page they cite studies have examined the co-occurrence of DP with topographic disorientation and dyslexia.
Response: Thanks for pointing this out. What we intended to point out in that sentence was that other neurodevelopmental disorders are typically not mentioned in exclusion or inclusion criteria for DP-studies, so we cannot get information about the frequencies of such comorbidities from looking at the literature. We have now rephrased this sentence to say this more clearly, and it now reads:
“Thus, with the exception of ASD, potential co-occuring neurodevelopmental disorders are typically not part of inclusion or exclusion criteria in studies of DP.”
- The extensive use of acronyms makes the study a bit hard to follow. I suggest reducing the number of acronyms used to just those that are well known (e.g. ADHD).
Response: We have now reduced the number of acronyms (e.g., deleting EFA, WQ, MA, SDAM, DAP-Q, CFPT, ARHQ) and only use those that are well known. For the sake of readability temporary acronyms for the scales are used in Table 6, and explained in a footnote to the table.
Method
- (3/120) Were subjects only included if they reported a lifelong impairment in face recognition?
Response: No that was not the case. While the PI 20 asks directly about this, it was not a part of the recruitment text. We have rephrased the sentence to avoid it sounding like a specific inclusion criteria, so it now reads:
“Participants were encouraged to participate if they experienced difficulties recognizing familiar faces in their everyday life, and thus suspected to suffer from DP”.
- (4/146-157) Is it possible that some of the participants did not select “yes” for some of the disorders they were asked about because they don’t know what they are? Was any further clarification provided as to what constitutes dyslexia, for example?
Response: That is a possibility that we cannot rule out. For object agnosia, aphantasia, synaesthesia and memory problems a short explanatory text was provided in brackets, as we assumed these to be more unfamiliar to the participants. What we hoped to achieve was to receive ‘Yes’-answers that indicated a medical diagnosis, as the question targeting more known developmental disorders was phrased “Do you have other diagnoses?” e.g. dyslexia, but then assuming that they were familiar with this term. As such, it is possible that more people would have responded yes had the questions been formulated differently.
- (4/167-171) Arizpe et al (2019) makes a strong case for not using the PI20 alone as a screening tool for prosopagnosia. I think the limitation of the PI20 being used in this way should at least be mentioned: https://link.springer.com/article/10.3758/s13428-018-01195-w.
Response: We now mention this explicitiy in section 5.2.2. Limitations:
“Some researchers regard self-reports of face recognition as highly valid due to moderate-high correlations (often ~.35-.55) between the PI20 (as well as other self-report measures of face recognition) and objective measures of face processing in typical individuals [70,124-126]. In contrast, Arizpe, et al. [127] conclude that the PI20 is not sensitive enough to be used as a screening tool alone, as done in the present study. Along the same vein , it has been shown that people in general have only modest insight into their own performance on objective measures of face recognition abilities [9,122,128], but that DPs have better insight into their objective performances in comparison [122].”
- I am happy to see that the authors report the validity metrics of the scales from which they used questions. However, I am not sure that the validity metrics for the entire scale, as reported on pages 4 & 5, can be used to justify the use of individual items from those scales.
Response: This is a good point. For the PI 20, we used the whole scale. For the other items, we wanted to at least select them from validated scales if at all possible, which is why we opted for the Wayfinding questionnaire and the Adult Reading History Questionnaire. Regarding the Wayfinding questionnaire, there are suggestions from the authors regarding clinical cut-offs for the individual items which we mention in the text. For the reading questionnaire, we now specifically note that “As we only used a subset of the items from this questionnaire, the psychometric properties of the included items is unknown.”
- Section 3.1.2: It would be useful if the authors added some evidence that the sample size they used is appropriate for the factor analysis.
Response: It is debated what the appropriate sample size for this type of factor analysis is. We now mention this specifically in section 5.2.2. Limitations:
“In addition, the current sample size of 115, and the additional metrics from the exploratory factor analysis, lies in the lower end to satisfy the criteria of Maccallum, et al. [132]. There are, however, differing views on the right sample size for this large sample procedure [133].”
- (6/284-286) When you refer to “a meaningful amount of description” are you referring to the individual level or the group level?
Response: Great question. We mean on the group level, when several participants experience a similar theme. However, as this is qualitative data, and given that the richness of the descriptions differ a lot, one individually rich and significant description could also count towards a meaningful amount of description.
- For determining those who have navigational problems, how did you decide to make the cutoff <=3?
Response: This is stated in section 2.4.3.: “De Rooij and colleagues [72] (p. 1051) define a score of ≤3 on an item as indicating clinically relevant complaints.”
- From the standpoint of making sure that there is representation of women in research studies, I find it concerning that biological sex or gender was not recorded.
Response: This is a good point, and a concern we generally take very seriously. The reason for not including gender / sex in the questionnaire was to preserve the anonymity of the respondents based on advice from our data protection officer.
Results
- Section 4.4 could benefit from some sort of summary of the contents of table 6 and 7, which could then reference the table.
Response: We have now added the following summary to the section 4.4.:
“Table 7 provides an overview of the diversity within the participants’ qualitative reports of their strengths and difficulties. A range of categories reflects domains where both impairments and strengths are reported (by different individuals). This applies to visual imagery, mathematical abilities, reading/spelling, memory, musicality, sports skills, and navigation, where, interestingly, some participants report severe problems while other report excelling. For some domains, e.g., imagery, reading/spelling, and navigation, more participants reported either strengths or difficulties, while for other domains (e.g., maths, musicality) representations of strengths and difficulties were more evenly distributed. For some categories, only impairment (object recognition, motor coordination, inattention), or abilities/strengths (synaesthesia, social interaction, visual arts) were represented in the qualitative categories.”
A summary of Table 6 is presented in section 4.3.
Discussion
- (15/423) What is “the counterpart” referring to?
Response: We have rephrased this sentence, so it now reads “For Aphantasia and Navigation Problems more participants reported impairment, while comparatively fewer reported performing well in these domains.”
- (16/470) I do not understand what this sentence is trying to say!
Response: Response: We have now rephrased this sentence to: “ADHD-like symptoms are generally not mentioned in the literature on DP, e.g., in these overviews [25,44], and to our knowledge there is only one published report of comorbid DP and ADHD [54].”.
- (16/490) You mention that aphantasia was reported frequently in both the qualitative and quantitative data. Could this be because participants were primed to consider aphantasia when completing the quantitative portion of the study?
Response: This is an interesting question, and we now mention this possibility in section 5.2.2. Limitations:
“There is also an important caveat in interpreting the frequencies with which specific disorders or deficits were reported, as we did pre-specify some of these in the questionnaire. Thus, had we included other specific disorders e.g., developmental coordination disorder, or OCD, our findings would likely look different. The mentioning of specific disorders in the questionnaire could also prime the participants to consider functions or disorders they would not have considered otherwise, e.g. aphantasia mentioned specifically and was frequently reported both in the quantitative and qualitative data”.
- (17/530) N=9/? And N=7/?
Response: These N’s refer to the number of participants in the cited studies. We have now deleted this information, given that it appeared confusing.
- (17/551-552) Object agnosia and “deficits in object recognition” seem, to me, to be different, with object agnosia being much more severe. Therefore, it seems fair to say that DP can occur both with and without impairments in object recognition (lines 545-546] but not that “object agnosia will present with some cases of DP.” I would argue that object agnosia, as an official diagnoses, would exclude a diagnosis of prosopagnosia (if you can’t identify basic level objects, then failure to recognize faces is likely secondary to that).
Response: Thanks for pointing this out. We have rephrased this so it now reads ‘deficits in object recognition are present in at least some cases of DP’, and have also substituted ‘object recognition problems’ for ‘object agnosia’ in other places where this was more appropriate.
- When relevant, and considering point #2 above, it is helpful to include the currently reported prevalence rates for each of the disorders examined in the discussion (as currently is the case for many, but not all). For example, do you know what percentage of the general population reports having at least one developmental disability?
Response: Regarding prevalence rates, we now comment in the manuscript on how these were estimated (see response to item 2). As for how many in the general population would report any of the impairments or disorders included in our study, we have not been able to find any literature providing indications of this. Most studies include estimates of intellectual and developmental disorders combined, and typically including disorders we did not specifically ask about like cerebral palsy or ASD. The closest we have found is an estimate from the The Behavior Risk Factor Surveillance System (BRFSS) in the US, where 22% answered ‘yes’ to one of the following two questions: (1) “Are you limited in any way in any activities because of physical, mental, or emotional problems?” (2) “Do you now have any health problem that requires you to use special equipment, such as a cane, a wheelchair, a special bed, or a special telephone?” (reference: https://doi.org/10.1016/j.dhjo.2014.11.002). But again –this measures disability far beyond the issues addressed in our study (and perhaps many of our participants would not indicate that they are ‘limited in any way in any activities’?). So, while the suggestion that we look for such an estimate is great, we have not been able to find a reasonable number to compare our data to.
Round 2
Reviewer 2 Report
While I appreciate the authors' changes to the manuscript, I continue to have an issue with the term "self-reported developmental prosopagnosia". It is more appropriate to call these people "individuals with self-reported face recognition deficits", especially considering that less than half (as the authors admit) would be likely to be classified as having developmental prosopagnosia. Using the term self-reported developmental prosopagnosia is misleading and could be harmful to the field. The field of developmental prosopagnosia has repeatedly warned against diagnosing DP using only self-reports (Arizpe, 2019; Corrow, Barton, 2016; Duchaine, 2008) and has reached a consensus that developmental prosopagnosia diagnosis requires self-report and objective evidence of face recognition deficits. Using the term "self-reported developmental prosopagnosia" in the title and throughout the manuscript will likely hurt the efforts towards getting all developmental prosopagnosia researchers to speak the same language and have comparable diagnostic results.
The second important remaining issue is the lack of a comparison group. Though I appreciate the authors including this in the limitations section, I don't think that's adequate. With the lack of a comparison group, the authors do not have much to respond to the critique that 'these are the comorbidities of people recruited in this method from the web at this time that may or may not have to do with face recognition complaints.' One potential way to demonstrate some specificity to those with self-reported face recognition deficits is to take the more severe PI20 group and compare it to the moderate/mild PI20 group. If there is significantly more of a comorbidity in the severe group (presumably who are more likely to have DP) compared to the mild/moderate group (less likely to have DP), than that would provide some evidence that the results presented are related to the severity of self-reported face recognition deficits.
Author Response
Please see the attached response letter.

Reviewer 3 Report
I would like to thank the authors for thought responses to my comments. While I still feel the study would benefit from a control group, I still see the value in a descriptive study of this kind at this point in the exploration of this question. I have no additional comments.
Author Response
Thank you for your comments on our paper and seeing the value in the study in spite of its limitations.